# Design of a Bacteriophage Cocktail Active against *Shigella* Species and Testing of Its Therapeutic Potential in *Galleria mellonella*

**DOI:** 10.3390/antibiotics11111659

**Published:** 2022-11-19

**Authors:** Andrey A. Filippov, Wanwen Su, Kirill V. Sergueev, Richard T. Kevorkian, Erik C. Snesrud, Apichai Srijan, Yunxiu He, Derrick E. Fouts, Woradee Lurchachaiwong, Patrick T. McGann, Damon W. Ellison, Brett E. Swierczewski, Mikeljon P. Nikolich

**Affiliations:** 1Wound Infections Department, Bacterial Diseases Branch, Walter Reed Army Institute of Research, Silver Spring, MD 20910, USA; 2Multidrug-Resistant Organism Repository and Surveillance Network, Bacterial Diseases Branch, Walter Reed Army Institute of Research, Silver Spring, MD 20910, USA; 3Department of Enteric Diseases, Armed Forces Research Institute of Medical Sciences, Bangkok 10400, Thailand; 4J. Craig Venter Institute, Rockville, MD 20850, USA; 5Bacterial Diseases Branch, Walter Reed Army Institute of Research, Silver Spring, MD 20910, USA

**Keywords:** *Shigella*, bacteriophage, phage therapeutics, *Galleria mellonella* infection model

## Abstract

Shigellosis is a leading global cause of diarrheal disease and travelers’ diarrhea now being complicated by the dissemination of antibiotic resistance, necessitating the development of alternative antibacterials such as therapeutic bacteriophages (phages). Phages with lytic activity against *Shigella* strains were isolated from sewage. The genomes of 32 phages were sequenced, and based on genomic comparisons belong to seven taxonomic genera: *Teetrevirus*, *Teseptimavirus*, *Kayfunavirus*, *Tequatrovirus*, *Mooglevirus*, *Mosigvirus* and *Hanrivervirus*. Phage host ranges were determined with a diverse panel of 95 clinical isolates of *Shigella* from Southeast Asia and other geographic regions, representing different species and serotypes. Three-phage mixtures were designed, with one possessing lytic activity against 89% of the strain panel. This cocktail exhibited lytic activity against 100% of *S. sonnei* isolates, 97.2% of *S. flexneri* (multiple serotypes) and 100% of *S. dysenteriae* serotypes 1 and 2. Another 3-phage cocktail composed of two myophages and one podophage showed both a broad host range and the ability to completely sterilize liquid culture of a model virulent strain *S. flexneri* 2457T. In a *Galleria mellonella* model of lethal infection with *S. flexneri* 2457T, this 3-phage cocktail provided a significant increase in survival.

## 1. Introduction

Shigellosis, or bacillary dysentery, is a disease caused by invasion of the colonic, rectal and distal ileal epithelium by *Shigella* spp. Shigellosis is a leading cause of diarrheal disease worldwide, particularly in developing countries [1,2], and is a continuing problem for civilian and military travelers visiting endemic regions [3,4]. Vaccine development [5,6] and other prophylactic measures [1] remain a high priority given the disease burden [7], increasing antibiotic resistance [8,9], and gaining appreciation of the post-infectious sequelae associated with shigellosis [10]. *Shigella* encompasses four species subdivided into serotypes and subserotypes, *Shigella dysenteriae* (15 serotypes and 2 provisional serotypes), *Shigella flexneri* (7 serotypes and 15 subserotypes), *Shigella sonnei* (one serotype), and *Shigella boydii* (20 serotypes) [8,11]. *S. sonnei* is the most common species found in high-income countries [1]. *S. flexneri* accounts for 30–60% of shigellosis cases in developing regions, necessitating coverage of prevalent *S. flexneri* serotypes in a multivalent *Shigella* vaccine or therapeutic [12]. Data from studies where culture-independent diagnosis was assessed, such as quantitative polymerase chain reaction (qPCR) for *Shigella*, indicate that traditional culture-based methods significantly underestimate the global burden of *Shigella*-associated illness [13,14]. Estimates from the Global Enteric Multicenter Study (GEMS) found that analysis using qPCR resulted in a 2 to 2.5 fold increase in the attributable fraction of *Shigella*-associated moderate-severe diarrheal disease [13,15].

Historically, diarrhea has been the most common illness reported by U.S. military service members during numerous military exercises and mobilizations to regions/theaters where sanitation conditions were poor [16,17]. *Shigella* and enterotoxigenic *Escherichia coli* (ETEC) are major bacterial etiological agents of travelers’ diarrhea [3,4]. Additionally, antimicrobial resistance to common antibiotics used for the treatment of travelers’ diarrhea, including ciprofloxacin, is significantly increasing in ETEC and *Shigella* isolates, especially from countries in Africa and South and Southeast Asia [8,18,19]. Thus, prophylaxis of shigellosis and ETEC infection among military and civilian contingents is a priority, since no licensed vaccine is available [5,6].

Bacteriophages (phages) have shown therapeutic efficacy against various multidrug-resistant infections in laboratory animals and humans, including individual compassionate use and several promising clinical trials [20,21]. A single lytic phage prevented *S. flexneri* adherence and invasion *in vitro*, using a cultured human colorectal adenocarcinoma cell model [22]. A cocktail of five ATCC phages was able to lyse 62/65 (95%) of *Shigella* strains that belong to all four species and oral gavage of mice with this cocktail shortly before and/or after *S. sonnei* infection significantly reduced bacterial burden in fecal and cecum samples and did not distort gut microbiota [23]. Similar phage effects were observed in mice treated with phages against oral *Listeria monocytogenes* [24] and *E. coli* O157:H7 infections [25]. In addition, oral administration of a cocktail of three phages prevented *Vibrio cholerae* infection in infant mice and rabbits [26].

The oral administration of phages was used in different countries for treatment against dysentery in humans since 1919 and appeared to result in more frequent positive outcomes than in patients who did not receive phage (for reviews, see [27,28]). Two double-blind placebo-controlled clinical trials were conducted in the USSR, designed according to contemporary WHO standards and enrolling thousands of subjects. In these trials, a 2.5-fold and 3.9-fold reduction in the prevalence of dysentery was observed in groups of children receiving a lyophilized and pectinized mixture of *Shigella*-specific lytic phages once every seven [29] or every three days [30], compared with control groups receiving a placebo.

The deliberate rational design of phage cocktails for prophylactic and/or therapeutic use is needed to ensure these antimicrobials will work well in concert with standard of care antibiotic treatments, and also to overcome the bacterial resistance that emerges naturally [31]. Phages are isolated from the environment and selected as candidates for use in cocktails based on attributes including burst size, host range within the diversity of the target pathogen, anti-biofilm activity, synergies with treatment antibiotics, and selection of host receptor diversity [20]. The main aim of this work was to initiate the rational design of phage cocktails against target *Shigella* species and serotypes for the development of antimicrobials with activity against emerging multidrug-resistant variants. Toward this aim, we isolated new lytic phages, designed initial 3-phage prototype cocktails with broad host range against key *Shigella* pathogens, and tested the efficacy of a lead cocktail in protecting against *S. flexneri* infection in a *Galleria mellonella* model.

## 2. Results

### 2.1. Phage Isolation

Using the eight *Shigella* strains selected for phage enrichment (Table 1), three fractions of Washington DC wastewater collected before any chemical treatment (Materials and Methods) yielded a high prevalence of phages with lytic activity against *Shigella*. Thirty-two phages were isolated from the sewage samples that were distinct and diverse based on the EcoRV restriction digestion patterns of their genomic DNA, each with its own unique restriction profile (data not shown).

### 2.2. Phage Genome Analysis

All 32 phages were sequenced, with genome sizes ranging from 38,701 to 170,646 bp (Table 2). Based on nucleotide BLAST analysis, these phages were classified into seven genera, *Tequatrovirus* (ESh16-18, 24-26, 28-36), *Mosigvirus* (ESh15, ESh27), *Teetrevirus* (ESh7-12), *Teseptimavirus* (ESh1-3, 6), *Kayfunavirus* (ESh23), *Mooglevirus* (ESh19-22), and *Hanrivervirus* (ESh4) within four families (*Straboviridae*, *Autographiviridae*, *Myoviridae* and *Drexlerviridae*) (See Table 2). An average nucleotide distance phylogenetic tree of these *Shigella* phages, based on whole genome sequences, is presented in Figure 1. Bioinformatic analysis indicated that these phage genomes do not contain genes that are potentially deleterious for phage therapeutic application, such as putative determinants of transduction, genes encoding antimicrobial resistance, toxins and other virulence factors. While all of the phages appear to be strictly lytic, four of them (ESh19-22) belong to subfamily *Ounavirinae* within the former family *Myoviridae*. Some representatives of this subfamily called “superspreaders” have been found to efficiently release intact plasmid DNA upon lysis and thus to stimulate horizontal gene transfer by transformation [32]. For example, the genome of phage ESh19 showed 84% identity at the nucleotide level with superspreaders SUSP1 and SUSP2 [32] using BLAST analysis. Therefore, before using them as therapeutics, phages ESh19-22 should be tested to exclude the ability to enhance gene transfer.

**Table 1 antibiotics-11-01659-t001:** *Shigella* strains used in this work.

#	Strain	Serotype	Origin	#	Strain	Serotype	Origin
**1**	*S. flexneri* 27	1	Vietnam	**50**	*S. flexneri* 83	3a	Kenya
**2**	*S. flexneri* 46	1	Thailand	**51**	*S. flexneri* 84	3a	Kenya
**3**	*S. flexneri* 13	1a	Bhutan	**52**	*S. flexneri* J17B *	3a	Japan
**4**	*S. flexneri* 28	1a	Vietnam	**53**	*S. flexneri* 38	3b	Vietnam
**5**	*S. flexneri* 61	1a	Nepal	**54**	*S. flexneri* 9	4	Cambodia
**6**	*S. flexneri* 2	1b	Cambodia	**55**	*S. flexneri* 22	4	Bhutan
**7**	*S. flexneri* 3	1b	Cambodia	**56**	*S. flexneri* 39	4	Vietnam
**8**	*S. flexneri* 14	1b	Bhutan	**57**	*S. flexneri* 54	4	Thailand
**9**	*S. flexneri* 15	1b	Bhutan	**58**	*S. flexneri* 70	4	Nepal
**10**	*S. flexneri* 19	1b	Bhutan	**59**	*S. flexneri* 40	4a	Vietnam
**11**	*S. flexneri* 29	1b	Vietnam	**60**	*S. flexneri* 41	4a	Vietnam
**12**	*S. flexneri* 30	1b	Vietnam	**61**	*S. flexneri* 55	5	Thailand
**13**	*S. flexneri* 47	1b	Thailand	**62**	*S. flexneri* M90T	5a	USA
**14**	*S. flexneri* 62	1b	Nepal	**63**	*S. flexneri* M90T55 ^b^*	5a	Laboratory
**15**	*S. flexneri* 82	1b	Kenya	**64**	*S. flexneri* 10	6	Cambodia
**16**	*S. flexneri* 63	1c	Nepal	**65**	*S. flexneri* 11	6	Cambodia
**17**	*S. flexneri* 16	2	Bhutan	**66**	*S. flexneri* 23	6	Bhutan
**18**	*S. flexneri* 31	2	Vietnam	**67**	*S. flexneri* 24	6	Bhutan
**19**	*S. flexneri* 48	2	Thailand	**68**	*S. flexneri* 42	6	Vietnam
**20**	*S. flexneri* 4	2a	Cambodia	**69**	*S. flexneri* 43	6	Vietnam
**21**	*S. flexneri* 5	2a	Cambodia	**70**	*S. flexneri* 56	6	Thailand
**22**	*S. flexneri* 17	2a	Bhutan	**71**	*S. flexneri* 57	6	Thailand
**23**	*S. flexneri* 18	2a	Bhutan	**72**	*S. flexneri* 71	6	Nepal
**24**	*S. flexneri* 32	2a	Vietnam	**73**	*S. flexneri* 72	6	Nepal
**25**	*S. flexneri* 33	2a	Vietnam	**74**	*S. flexneri* 85	6	Kenya
**26**	*S. flexneri* 49	2a	Thailand	**75**	*S. flexneri* SSU2415 *	6	USA
**27**	*S. flexneri* 50	2a	Thailand	**76**	*S. flexneri* CCH060 *	6	Unknown
**28**	*S. flexneri* 64	2a	Nepal	**77**	*S. flexneri* 58	var. X	Thailand
**29**	*S. flexneri* 65	2a	Nepal	**78**	*S. flexneri* 44	var. Y	Vietnam
**30**	*S. flexneri* 81	2a	Kenya	**79**	*S. sonnei* 1	NA	Cambodia
**31**	*S. flexneri* 2457T *	2a	Japan	**80**	*S. sonnei* 12	NA	Bhutan
**32**	*S. flexneri* BS103 ^a^*	2a	Laboratory	**81**	*S. sonnei* 26	NA	Vietnam
**33**	*S. flexneri* 6	2b	Cambodia	**82**	*S. sonnei* 45	NA	Thailand
**34**	*S. flexneri* 34	2b	Vietnam	**83**	*S. sonnei* 60	NA	Nepal
**35**	*S. flexneri* 66	2b	Nepal	**84**	*S. sonnei* Moseley *	NA	USA
**36**	*S. flexneri* ATCC 12022	2b	Unknown	**85**	*S. sonnei* ATCC 25931	NA	Panama
**37**	*S. flexneri* 35	2ab	Vietnam	**86**	*S. dysenteriae* 59	1	Thailand
**38**	*S. flexneri* 51	3	Thailand	**87**	*S. dysenteriae* 73	1	Nepal
**39**	*S. flexneri* 7	3a	Cambodia	**88**	*S. dysenteriae* 1617 *	1	Guatemala
**40**	*S. flexneri* 8	3a	Cambodia	**89**	*S. dysenteriae* 74	2	Nepal
**41**	*S. flexneri* 20	3a	Bhutan	**90**	*S. dysenteriae* 75	9	Nepal
**42**	*S. flexneri* 21	3a	Bhutan	**91**	*S. dysenteriae* 76	12	Nepal
**43**	*S. flexneri* 36	3a	Vietnam	**92**	*S. dysenteriae* 87	12	Kenya
**44**	*S. flexneri* 37	3a	Vietnam	**93**	*S. boydii* 77	1	Nepal
**45**	*S. flexneri* 52	3a	Thailand	**94**	*S. boydii* 25	2	Bhutan
**46**	*S. flexneri* 53	3a	Thailand	**95**	*S. boydii* 78	2	Nepal
**47**	*S. flexneri* 67	3a	Nepal	**96**	*S. boydii* 86	2	Kenya
**48**	*S. flexneri* 68	3a	Nepal	**97**	*S. boydii* 79	10	Nepal
**49**	*S. flexneri* 69	3a	Nepal	**98**	*S. boydii* 80	12	Nepal

^a^ Non-invasive plasmid-cured strain of 2457T [33]; ^b^ non-invasive plasmid-cured strain of M90T [34]; * strain selected for phage enrichments. NA, not applicable.

**Table 2 antibiotics-11-01659-t002:** The characteristics of 32 *Shigella* phages isolated in this study.

Phage ID	Genome Size, bp	Accession No.	Phage taxonomy ^a^			Closest Relative in NCBI Database ^b^	
			Family	Subfamily	Genus	Definition	Accession No.
**ESh1**	39,034	ON528715	*Autographiviridae*	*Studiervirinae*	*Teseptimavirus*	64795_ec1	KU927499
**ESh2**	39,818	ON528716	*Autographiviridae*	*Studiervirinae*	*Teseptimavirus*	JeanTinguely Bas64	MZ501081
**ESh3**	39,180	ON528717	*Autographiviridae*	*Studiervirinae*	*Teseptimavirus*	64795_ec1	KU927499
**ESh4**	51,077	ON528718	*Drexlerviridae*	*Tempevirinae*	*Hanrivervirus*	herni	NC_049823
**ESh6**	39,381	ON528719	*Autographiviridae*	*Studiervirinae*	*Teseptimavirus*	JeanTinguely Bas64	MZ501081
**ESh7**	39,724	ON528720	*Autographiviridae*	*Studiervirinae*	*Teetrevirus*	vB_KpnP_IME305	OK149215
**ESh8**	38,701	ON528721	*Autographiviridae*	*Studiervirinae*	*Teetrevirus*	phiYe-F10	NC_047755
**ESh9**	39,308	ON528722	*Autographiviridae*	*Studiervirinae*	*Teetrevirus*	2050H2	NC_047844
**ESh10**	38,729	ON528723	*Autographiviridae*	*Studiervirinae*	*Teetrevirus*	vB_YenP_AP5	KM253764
**ESh12**	39,704	ON528724	*Autographiviridae*	*Studiervirinae*	*Teetrevirus*	2050H2	NC_047844
**ESh15**	168,076	ON528725	*Straboviridae*	*Tevenvirinae*	*Mosigvirus*	SHSML-52-1	KX130865
**ESh16**	165,784	ON528726	*Straboviridae*	*Tevenvirinae*	*Tequatrovirus*	Sfk20	MW341595
**ESh17**	166,355	ON528727	*Straboviridae*	*Tevenvirinae*	*Tequatrovirus*	slur07	LN881732
**ESh18**	165,470	ON528728	*Straboviridae*	*Tevenvirinae*	*Tequatrovirus*	Kha5h	NC_054905
**ESh19**	87,867	ON528729	*Myoviridae* ^c^	*Ounavirinae*	*Mooglevirus*	vB_EcoM_3HA14	MN342151
**ESh20**	89,515	ON528730	*Myoviridae* ^c^	*Ounavirinae*	*Mooglevirus*	vB_EcoM_3HA14	MN342151
**ESh21**	86,414	ON528731	*Myoviridae* ^c^	*Ounavirinae*	*Mooglevirus*	KPS64	MK562502
**ESh22**	88,154	ON528732	*Myoviridae* ^c^	*Ounavirinae*	*Mooglevirus*	vB_EcoM_3HA14	MN342151
**ESh23**	40,156	ON528733	*Autographiviridae*	*Studiervirinae*	*Kayfunavirus*	SFPH2	NC_048025
**ESh24**	167,086	ON528734	*Straboviridae*	*Tevenvirinae*	*Tequatrovirus*	vB_EcoM_F1	NC_054912
**ESh25**	166,499	ON528735	*Straboviridae*	*Tevenvirinae*	*Tequatrovirus*	Aplg8	NC_054902
**ESh26**	167,539	ON528736	*Straboviridae*	*Tevenvirinae*	*Tequatrovirus*	UGKSEcP2	OV876900
**ESh27**	168,955	ON528737	*Straboviridae*	*Tevenvirinae*	*Mosigvirus*	phiC120	NC_055718
**ESh28**	164,289	ON528738	*Straboviridae*	*Tevenvirinae*	*Tequatrovirus*	JK23	MK962752
**ESh29**	166,160	ON528739	*Straboviridae*	*Tevenvirinae*	*Tequatrovirus*	vB_EcoM_Shinka	MZ502379
**ESh30**	170,189	ON528740	*Straboviridae*	*Tevenvirinae*	*Tequatrovirus*	fPS-2	NC_054943
**ESh31**	167,224	ON528741	*Straboviridae*	*Tevenvirinae*	*Tequatrovirus*	PhiZZ30	NC_054938
**ESh32**	169,173	ON528742	*Straboviridae*	*Tevenvirinae*	*Tequatrovirus*	vB_EcoM_G2133	MK327928
**ESh33**	166,484	ON528743	*Straboviridae*	*Tevenvirinae*	*Tequatrovirus*	vB_SboM_Phaginator	OL615012
**ESh34**	167,055	ON528744	*Straboviridae*	*Tevenvirinae*	*Tequatrovirus*	Sfk20	MW341595
**ESh35**	166,919	ON528745	*Straboviridae*	*Tevenvirinae*	*Tequatrovirus*	KIT03	NC_054923
**ESh36**	170,646	ON528746	*Straboviridae*	*Tevenvirinae*	*Tequatrovirus*	T4_ev151	LR597660

^a^ General taxonomy for all phages: *Viruses*, *Duplodnaviria* (realm), *Heunggongvirae* (kingdom), *Uroviricota* (phylum), *Caudoviricetes* (class), *Caudovirales* (order), then families, subfamilies and genera as indicated in Table 2. ^b^ National Center for Biotechnology Information. ^c^ The family assignment of subfamily *Ounavirinae* viruses is unclear in the current taxonomy, so the previous family assignment is retained herein.

### 2.3. Phage Morphology

The morphology of phage virions was studied using transmission electron microscopy (Figure 2). Phage particles exhibited typical morphology associated with their family classification: myovirus phages with long contractile tails in genera *Tequatrovirus*, *Mosigvirus* (both now reclassified from *Myoviridae* to family *Straboviridae*) and *Mooglevirus* (family *Myoviridae*); podophages in genera *Teetrevirus*, *Teseptimavirus* and *Kayfunavirus* with short non-contractile tails (family *Autographiviridae*); and a siphophage with long non-contractile tail in the genus *Hanrivervirus* (family *Drexlerviridae*). Virion morphologies were consistent with what was expected based on the morphologies of phages with similar genome sequences.

### 2.4. Prototype Phage Cocktails

First, a panel of 12 phages was selected from the larger collection for further testing based on breadth of lytic host range across bacterial strains that represented the chief *Shigella* serotypes being targeted (shown in Table 3). Four mixtures or cocktails consisting of three phages each were then developed based on the initial lytic properties of individual candidate phages. Three out of the four mixtures (##1, 2 and 4) demonstrated the ability to completely clarify and sterilize liquid cultures of *S. flexneri* 2a 2457T (Table 4).

### 2.5. Host Range Testing

Activities of the 12 selected phages and the four 3-phage cocktails were tested against a panel of 95 *Shigella* strains assembled by the Armed Forces Research Institute of Medical Sciences composed mainly of clinical isolates from Southeast Asia, but also from East Asia, Africa, South America and the USA (Table 5 and Appendix A). Overall, the 12 phages were able to lyse 86/95 (90.5%) of *Shigella* strains (Appendix A). All of the 3-phage cocktails showed broad host ranges and killed 100% of *S. sonnei* isolates, 82–97% of *S. flexneri* (including serotypes 1, 1a, 1b, 1c, 2, 2a, 2b, 3a, 3b, 4, 4a, 5, and 6, as well as X and Y variants) and 100% of *S. dysenteriae* serotypes 1 and 2 (Table 5). Since 89/95 (93.7%) of the strains were pigmented on Congo Red agar (data not shown) and thus carried the virulence plasmid, it appears that these phage mixtures successfully kill virulent strains of *Shigella*. The representatives of some *Shigella* groups were not susceptible to this initial collection of candidate therapeutic phages: *S. dysenteriae* serotypes 9 and 12 and most *S. boydii* isolates.

### 2.6. Phage Treatment of Shigella Infection of G. mellonella Larvae

Based on the plating efficiencies of phages ESh12, ESh18 and ESh29, their individual lytic activities, and also their combined killing effect in liquid culture and host range, prototype cocktail #4 was selected to test therapeutic efficacy in the wax moth (*G. mellonella*) larvae infection model as an initial, more rapid and economical *in vivo* assessment. The therapeutic effect using the individual component phages and the 3-phage mixture was tested in the treatment of *S. flexneri* strain 2457T infection of *G. mellonella* larvae. Administration of the phages and the cocktail at the doses used did not cause adverse effects on the larvae, a concern because of endotoxin carryover in phage purification (not shown). Survival was extended in larvae infected with a lethal dose of strain 2457T and treated 30 min later with either the individual phages or the cocktail (Figure 3). The rate of survival of the larvae after 72 h increased from 40–50% without phage treatment to 55–85% with treatment using the individual phages or the cocktail. Phage ESh29 alone (Figure 3c) or the 3-phage cocktail (Figure 3d) each provided an increase in survival to about 85%; this indicates that ESh29 provides the predominant therapeutic effect of the cocktail against strain 2457T, though all three component phages provided an increase in survival in this model (Figure 3). Higher doses of the three individual phages or the mixture did not correlate with higher survival after treatment, however (not shown).

## 3. Discussion

The high prevalence and severity of shigellosis, particularly in developing countries among both residents and travelers [1,4], and increasing drug resistance in *Shigella* spp. isolates [8] indicate that new antibacterials are needed to augment antibiotics. Phages are a promising option for the prophylaxis and therapy of shigellosis. Previous prophylactic oral treatment of children with a mixture of lytic phages resulted in a significant reduction of shigellosis in comparison with control groups receiving a placebo [29,30]. A 5-phage cocktail administered orally to mice before and/or after *S. sonnei* infection significantly reduced the numbers of bacteria in fecal and cecum specimens [23]. Optimization of a phage cocktail and more frequent phage application (perhaps every day or every other day) could potentially result in an even higher efficacy of shigellosis prophylaxis than what was observed in these studies. More research is required to isolate lytic phages of *Shigella* and characterize them *in vitro* and *in vivo* toward developing robust fixed phage cocktails to prevent and treat drug-resistant shigellosis, with potential benefits for civilian travelers and deployed military personnel.

The purpose of this work was to isolate a panel of lytic phages with broad activity against diverse *Shigella* strains, to develop prototype phage cocktails and evaluate the efficacy of phage treatment in a waxworm model. The eight strains of *Shigella* used for phage enrichment included *S. flexneri* (serotypes 2a, 3a, 5a, and 6), *S. sonnei*, and *S. dysenteriae* (serotype 1) (Table 1). Thirty-two lytic phages active against *Shigella* species (Table 2) were isolated from three fractions of Washington DC sewage (grit chamber water, secondary effluent and blend sludge) collected on the same day. Whole-genome sequencing and analysis enabled classification of the phages into seven viral genera (Table 2, Figure 1).

The virus family represented by the largest number of phages in the panel (17) was *Straboviridae* (formerly *Myoviridae*), including subfamily *Tevenvirinae*, genera *Tequatrovirus* (T4-like, 15 phages) and *Mosigvirus* (2 phages). T4-like phages are strictly lytic, do not show significant DNA sequence identity with bacterial genomes [37,38] and have broad host ranges among *Shigella* [23,39], pathogenic strains of *E. coli* [38,40,41,42], both *E. coli* and *Salmonella* [43,44], *E. coli*, *Salmonella* and *Shigella* [45], *Yersinia pestis* and *Yersinia pseudotuberculosis* [46], even *Acinetobacter baumannii* [47] and *Stenotrophomonas maltophilia* [48]. However, *S. maltophilia* T4-like phage DLP6 encodes a transposon that might stimulate gene transfer and thus is not a favorable candidate for phage therapy [48]. This suggests that genomes of even strictly lytic phages should be analyzed in depth to exclude potentially detrimental gene content before using them as therapeutics. *Mosigvirus* phages are similar to those belonging to the genus *Tequatrovirus* and were also considered T4-like phages until recently, when they were reclassified into a separate genus. They are also obligately lytic and demonstrate broad activity against *Shigella* [23,49] and pathogenic *E. coli* [41]. Four of the 32 phages belonged to subfamily *Ounavirinae* within family *Myoviridae*, genus *Mooglevirus*.

These 21 myoviral phages within *Straboviridae* and *Ounavirinae* isolated in this study did not show any significant DNA sequence similarity to genes encoding integrases, recombinases, transposases, excisionases, and repressors of the lytic cycle, nor to any bacterial genes, including drug resistance and pathogenicity determinants. The seventeen phages that belong to subfamily *Tevenvirinae* within *Straboviridae* appear to be safe for therapeutic application based on gene content. Subfamily *Ounavirinae* was named after diagnostic lytic *Salmonella* phage O1 (or Felix O1), proposed for therapy and control of *Salmonella* in food [50]. Although four *Ounavirinae* phages discovered by our team (ESh19-22) appear to be virulent, they share high genome identity with phages SUSP1 and SUSP2 (subfamily *Ounavirinae*, genus *Suspvirus*). SUSP1 and SUSP2, called “superspreaders,” have the demonstrated ability to efficiently release intact plasmid DNA upon lysis, followed by enhanced horizontal gene transfer via transformation [32]. Unless this ability can be experimentally excluded for ESh19-22, these four phages cannot be recommended for therapeutic use because of the risk of potentially spreading drug resistance or virulence determinants.

Ten of the phages were classified as members of three genera within family *Autographiviridae* (formerly *Podovoridae*) and subfamily *Studiervirinae*, including *Teetrevirus* (T3-like phages, ESh7-10 and ESh12), *Teseptimavirus* (T7-like phages, ESh1-3 and ESh6) and *Kayfunavirus* (ESh23). *Shigella* phages that belong to subfamily *Studiervirinae* appear to be relatively rare. For example, among 69 *Shigella* phages deposited in GenBank and listed in a recent review article by Subramanian et al. [51], there are 27 myophages of subfamily *Tevenvirinae* (18 of which are members of genus *Tequatrovirus*), while there is only one representative of podovirus subfamily *Studiervirinae*, *Kayfunavirus* phage SFPH2, and no T3- or T7-like phages. SFPH2 has been shown to lyse strains of *S. flexneri* 2, 2a and Y [52]. However, it was observed long ago that *E. coli* phages T3 and T7 are able to lyse some *S. sonnei* strains [53,54]. T3- and T7-like phages are virulent, have robust lytic activity [55,56,57], do not encode toxic proteins [58] and thus appear to be promising as candidate therapeutics. *Teetrevirus* phage KPP-5 exhibited a broad host range for *Klebsiella pneumoniae* strains [59] and *Teseptimavirus* phage EG1 was specific for uropathogenic isolates of *E. coli* [58]. Another T7-like phage, φA1122, is capable of lysing the vast majority of diverse *Y. pestis* strains [60]. Genomic analysis of all 10 podophages isolated in this study revealed no potentially detrimental genetic information, so these can be considered as candidate therapeutics. Finally, one phage, ESh4, belonged to family *Drexlerviridae* (formerly *Siphoviridae*), subfamily *Tempevirinae*, genus *Hanrivervirus*. The first representative of this genus, virulent phage pSf-1 isolated in Korea, showed lytic activity against *S. flexneri*, *S. boydii* and *S. sonnei* [61]. ESh4 also seems to be a candidate therapeutic phage because no potentially deleterious genes were found in its genome. Transmission electron microscopy confirmed that the 32 phages isolated in this work belong to myo-, podo- and siphoviruses (Figure 2).

Initial use of host range testing against a small *Shigella* strain panel allowed for the selection of 12 phages with broader activity for further characterization (Table 3). This selection included representatives of genera *Tequatrovirus* (ESh16-18, ESh29, ESh31, ESh33, and ESh35), *Mosigvirus* (ESh27), *Mooglevirus* (ESh22), *Teetrevirus* (ESh9 and ESh12), and *Teseptimavirus* (ESh1). Four 3-phage cocktails were developed from these phages and tested for the ability to lyse and kill *S. flexneri* 2a 2457T, a fully virulent challenge strain that has been used globally in animal trials as a virulent *Shigella* challenge [34]. Three of these cocktails (mixtures ##1, 2 and 4) were able to completely lyse and sterilize broth cultures of strain 2457T (Table 4), suggesting that these phage combinations successfully kill the entire bacterial population, including any mutants resistant to the individual phages. Bacterial resistance is developed less frequently to phage cocktails than to single phages because well designed cocktails usually contain phages that use different cell surface receptors, and mutants resistant to one phage can be lysed by other cocktail components [62]. These results also indicated that strain 2457T does not rapidly develop resistance to these three 3-phage cocktails.

The host ranges of the 12 selected phages and four 3-phage cocktails were evaluated using a diversity panel of 95 *Shigella* clinical strains isolated in Southeast Asia and East Asia, Africa, South America, and the USA (Table 5 and Appendix A). The lytic spectra of individual phages ranged from 59% to 87%, and altogether the 12 phages were able to lyse 86/95 (90.5%) of the *Shigella* strains listed in Appendix A. All four 3-phage cocktails had broad host ranges, with activity against 100% of *S. sonnei* isolates, 82–97% of *S. flexneri* (serotypes 1, 1a, 1b, 1c, 2, 2a, 2b, 3a, 3b, 4, 4a, 5, and 6, as well as X and Y variants) and 100% of *S. dysenteriae* serotypes 1 and 2 (Table 5). Therefore, they had killing activity against virtually all of the *Shigella* subtypes that cause travelers’ diarrhea in Southeast Asia. The lytic activity observed for mixture #15 against the entire *Shigella* diversity panel was 88.8%. Importantly, 94% of the strains (89/95) were pigmented on Congo Red agar and thus possessed the virulence plasmid, indicating the selected phages and phage cocktails can efficiently kill virulent *Shigella* strains. *S. dysenteriae* serotypes 9 and 12 and most *S. boydii* isolates were resistant to all of the phages tested. Our data on the broad host ranges of individual *Shigella* phages and their mixtures agree with the results of others, but the bacterial strain panel used in this study may be better characterized and more diverse. For example, commercially available INTESTI, PYO and Septaphage phage cocktail products manufactured in Georgia, respectively possessed lytic activity against 19/20 (95%), 19/20 and 11/20 (55%) strains of *Shigella* spp. isolated in Switzerland (species/serotype breakdown not provided) [63]. A cocktail of two T1-like phages was able to lyse 85% of MDR isolates of *S. sonnei* (44) and *S. flexneri* (26, without serotype breakdown) from different provinces of Iran [9]. A cocktail of five ATCC phages that belong to genera *Tequatrovirus* (3), *Mosigvirus* (1) and *Tequintavirus* (1; family *Demerecviridae*, subfamily *Markadamsvirinae*) provided a lytic effect against all strains of *S. sonnei* (18), *S. dysenteriae* (5), *S. boydii* (4), and 35/38 *S. flexneri* isolates (without serotype breakdown for the latter three species) [23].

Use of waxworm (*G. mellonella* larvae) infection models to study bacterial virulence and test new antimicrobials, including phages, offers low cost, technical simplicity and lack of ethical restrictions, in contrast to vertebrate models [64]. *G. mellonella* larvae have been used in *Shigella* virulence studies: injection of *S. flexneri* 2a 2457T was lethal for waxworms, while oral force feeding did not cause any death or clinical manifestations [65]. In this effort, we used a waxworm injection model to evaluate phage therapeutic efficacy against *S. flexneri* 2a 2457T infection. Prototype cocktail #4 was selected for this model based on the plating efficiencies of phages ESh12, ESh18 and ESh29 on strain 2457T, their individual lytic activities, the complete sterilization of liquid culture by the cocktail, and also host range of the mixture. Administration of each individual phage and the cocktail at the doses used did not cause adverse effects on the larvae, a concern because of endotoxin carryover in phage purification (not shown). Survival was significantly extended in larvae infected with a lethal dose of strain 2457T and treated 30 min later with the individual phages or the cocktail (Figure 3). The rate of survival of the larvae after 72 h increased from 40–50% without phage treatment to 55–85% with treatment using the individual phages or the cocktail (Figure 3). Both phage ESh29 alone and the 3-phage mixture provided an increase in survival to about 85%; this indicated that ESh29 may provide the majority of the therapeutic effect of the cocktail against strain 2457T, though each of the three component phages alone provided an increase in survival in this model (Figure 3). However, providing higher MOIs of the three individual phages or the mixture did not correlate with higher survival after treatment (not shown), perhaps because treatment efficacy was already saturating at the 1:1 MOI. The phage therapeutic effects observed in this study are comparable with those observed by others for lytic phages used in waxworms infected with *Burkholderia cepacia* [66], *Clostridium difficile* [67], *K. pneumoniae* [68], vancomycin-resistant *Enterococcus faecium* [69], and methicillin-resistant *Staphylococcus aureus* [70], though in the latter case, the observed phage effect was dose-dependent. One group first showed phage efficacy against *Pseudomonas aeruginosa* and *A. baumannii* infections in *Galleria* and then confirmed it in a mouse acute pneumonia model to indicate the relevance of the *Galleria* model [71,72].

The initial stages of rational phage cocktail design were employed in this study, with a focus on component phage host range and cocktail lytic activity. This effort built candidate phage mixtures for preclinical testing and potential development for prophylaxis and treatment of the common pathogens that cause shigellosis, a significant medical problem for large populations in developing countries, deployed military service members, and travelers. Initially we had anticipated the need to design different cocktails against representatives of the predominant *Shigella* species and serotypes circulating regionally in different parts of the world, but this initial effort indicated that it may well be possible to address the key *Shigella* pathogens on a global level using a single phage cocktail formulation, if well designed. A treatment effect against *S. flexneri* 2a was demonstrated here in the *Galleria* model, but further work must be done to determine whether such phage cocktails are also effective in treating *Shigella* infections in relevant mammalian models, and eventually in humans. Prophylaxis may be the main role for phages if they have a limited effect on human disease once underway because of the *Shigella* invasive and intracellular lifestyle [1]. Additionally, it will be necessary to demonstrate that the *in vitro* lytic spectrum of the phage cocktail translates to breadth of activity *in vivo*. The *Galleria* model can be used to test the efficacy of treatment against the main serotypes of *S. flexneri* and *S. sonnei* that cause the majority of disease and also allow for testing against a greater variety of pathogen strains than is feasible using mammalian models.

## 4. Materials and Methods

### 4.1. Bacterial Strains and Culture Media

The *Shigella* strains used in this work are presented in Table 1 and Appendix A. Bacteria were grown in Heart Infusion Broth (HIB, Becton, Dickinson and Co., Franklin Lakes, NJ, USA) or on 1.5% HIB agar plates at 37 °C. Semisolid 0.7% HIB agar was used as overlay for phage plating [73].

### 4.2. Isolation of Phages

Wastewater samples from three different sites and reactors (grit chamber water, secondary effluent and blend sludge) within the Blue Plains Wastewater Treatment Plant (Washington DC) were used as source materials for phage isolation. Eight strains were used for phage enrichment: *S. flexneri* 2a 2457T (virulent model strain broadly used in *in vitro* and *in vivo* studies [74]), BS103 (avirulent derivative of 2457T [75]), *S. flexneri* 3a J17B (wild type *S. flexneri* 3a strain from the Walter Reed Army Institute of Research/WRAIR collection [74]), *S. flexneri* 5a M90T55 (avirulent derivative of M90T [76]), *S. flexneri* 6 SSU2515 (wild type *S. flexneri* 6 strain from WRAIR collection; the source was a sheep outbreak in Florida in 1973 [Malabi Venkatesan, personal communication]), *S. flexneri* 6 CCH060 (wild type *S. flexneri* 6 strain from WRAIR collection [74]), *S. sonnei* Moseley (type strain from WRAIR collection [77]), and *S. dysenteriae* 1 1617 (wild type *S. dysenteriae* 1 from WRAIR collection [78]) (Table 1). The wastewater samples were centrifuged for 60 min at 5500× *g*. Centrifugation was repeated for blend sludge. Grit chamber water and secondary effluent supernatants were filtered using sterile 0.22-μm filters, while the blend sludge supernatant was consecutively filtered through 0.8-μm, 0.45-μm and 0.22-μm filters (MilliporeSigma, Bollington, MA, USA). Then, the samples were processed as previously described [79]. Briefly, 5× HIB was mixed with each filter-sterilized sample at a 1:5 ratio, and overnight broth culture of each enrichment *Shigella* strain was added to create 24 enrichment mixtures (three wastewater fractions by eight enrichment strains). The enrichment mixtures were incubated overnight at 37 °C with shaking at 200 rpm, and the supernatant was sterilized using a 0.22-μm filter. The resulting lysates were evaluated for plaque formation on double-layer HIB agar plates [73]. Phage purification was performed by three rounds of single plaque isolation.

### 4.3. Phage Propagation

Bacteriophages were propagated on *S. flexneri* strain BS103. The host bacteria were grown in HIB supplemented with 5 mM calcium chloride and incubated at 37 °C with shaking at 200 rpm. Phage lysate was added to 250 mL of an early exponential phase bacterial culture grown in HIB (OD_600_ of 0.1–0.2) at a multiplicity of infection (MOI) of 0.01–0.1 and incubated in a 500-mL plastic Erlenmeyer flask at 37 °C and 200 rpm until visible lysis occurred. Phage lysate was treated with chloroform (5%, vol./vol.). Bacterial debris was removed by centrifugation for 15 min at 5500× *g*. Phage particles from the supernatant were concentrated by centrifugation for 3 h at 13,250× *g*. Phage pellets were resuspended in 1/40 vol. of SM buffer (Teknova, Hollister, CA, USA). Bacterial debris was removed by centrifugation for 15 min at 5500× *g*, supernatant was collected and filtered through a sterile 0.22-μm PVDF membrane (MilliporeSigma, Sigma-Aldrich Inc., Saint Louis, MO, USA). Endotoxin levels in phage suspensions were tested with the Endosafe nexgen-PTS device (Charles River Laboratories, Wilmington, MA, USA), and if needed, further purified using EndoTrap bulk resin (Hyglos GmbH, Bernried am Starnberger See, Germany) according to the manufacturer’s protocol, to ensure that the endotoxin level was below 500 EU per 10^9^ PFU (plaque-forming units), approximating the U.S. Food & Drug Administration guidance for human use.

### 4.4. Phage DNA Isolation, Restriction Analysis and Genome Sequencing

Phage genomic DNA was extracted as described previously [80]. To identify unique phages, DNA samples were treated with restriction endonuclease EcoRV (New England BioLabs Inc., Ipswich, MA, USA) according to the manufacturer’s protocol and DNA fragments were separated using agarose gel electrophoresis. Phage genomic DNA of 32 phages with unique restriction profiles was sequenced on a MiSeq benchtop sequencer (Illumina, Inc., San Diego, CA, USA). Libraries were constructed using the Kapa HyperPlus library preparation kit (Roche Diagnostics, Indianapolis, IN, USA). Libraries were quantified using the Kapa library quantification kit Illumina/Bio-Rad iCycler (Roche Diagnostics) on a CFX96 real-time cycler (Bio-Rad, Hercules, CA, USA). For the MiSeq, libraries were normalized to 2 nM, pooled, denatured, and diluted to 20 pM. The pooled samples were further diluted to a final concentration of 14 pM. Samples were sequenced using MiSeq reagent kit v3 (Illumina; 600 cycles; 2 × 300 bp). Short-read sequencing data were trimmed for adapter sequence content and quality using Btrim64. Overlapping sequence reads were merged using FLASH. De novo assembly was performed using Newbler (v2.7). Minimum thresholds for contig size and coverage were set at 200 bp and 49.5×, respectively. The annotation of open reading frames and sequence similarity searches were performed as described earlier [79].

### 4.5. Phage Phylogenetic Tree

A phage whole genome phylogeny was generated from an ANI (Average Nucleotide Identity)-based distance matrix calculated with the Mash program [35] as described previously [81]. Briefly, a sketch file was created from the 32 described *Shigella* phage genomes isolated and sequenced in this study, plus 25 obtained from GenBank (11 *Escherichia*, 6 *Shigella*, 3 *Yersinia*, 2 *Serratia*, 1 *Klebsiella* and 2 *Enterobacteria* phages) with BLASTN [Basic Local Alignment Search Tool searching the NCBI nucleotide database] matches to the *Shigella* phages), with 5000 13mers generated per genome (i.e., mash sketch -k 13 -s 5000). The sketch file was then compared to all the phage genome sequences to generate the ANI matrix using the Mash distance command using default settings. The Gaussian Genome Representative Selector with Prioritization (GGRaSP) [36] R-package was used to calculate the UPGMA (Unweighted Pair Group Method with Arithmetic Mean) phylogeny from the ANI distance matrix, using GGRaSP options -e 5 -d 100 -m average. The resulting dendrogram was translated into Newick format within GGRaSP using the APE R package [82], loaded into the iTOL tree viewer [83], and annotated with taxonomic information.

### 4.6. Transmission Electron Microscopy

Phages were prepared for transmission electron microscopy as described previously [84], with minor modifications. Briefly, phage suspensions were washed twice with 0.1% ammonium acetate using centrifugation for 3 h at 13,250× *g* and phage titers were adjusted to 10^9^ PFU/mL. Phage particles were deposited on 300 mesh carbon-coated copper grids (Electron Microscopy Sciences, Hatfield, PA, USA), stained with 2% uranyl acetate for 1 min and examined in a JEOL JEM-1400 electron microscope at 80 kV. Images were analyzed with Image J software v. 1.53 (National Institutes of Health, Bethesda, MD, USA).

### 4.7. Phage Host Range Testing

Phage host ranges were determined using a micro-spot plating assay [80]. Briefly, 10-fold serial phage dilutions were prepared in a sterile flat-bottomed 96-well plate. A 2-μL aliquot of each phage dilution, ranging from 10^−1^ to 10^−8^, was spotted with a multichannel pipette on 0.7% HIB agar overlay infused with *Shigella* culture and incubated overnight at 37 °C. The morphology of individual plaques was evaluated and the results were scored from 4+ to 0 as follows: 4+, totally clear spots, isolated large clear plaques in the highest phage dilutions (highly positive result); 3+, clear spots, clear plaques of medium or small size, or large turbid plaques (positive result); 2+, clear or turbid spots, tiny clear or turbid plaques, sometimes barely countable (slightly positive result); 1+, lysis from without indicated by very faint, turbid spots or clear spots in first dilutions, no plaques (negative result); 0, no lysis spots or plaques (negative result).

### 4.8. Assessment of Phage Protection against Infection of G. mellonella Larvae with Shigella Strains

To determine if phages would provide an *in vivo* therapeutic effect against *Shigella*, a *G. mellonella* larva (wax worm) model of infection was utilized [72]. *S. flexneri* 2457T was grown to exponential phase, washed and resuspended in PBS to approximately 1 × 10^7^ CFU per mL. Waxworms (Vanderhorst, Inc., St. Marys, OH, USA) in the final-instar larval stage and weighing 200–300 mg were saved and housed in clean plastic Petri dishes, 10 worms per group. Worms were inoculated with 10 μL of the bacterial suspension prepared above into the last left proleg using a 300-μL BD Insulin syringe (Becton Dickinson, 1 Becton Drive, Franklin Lakes, NJ, USA). After 30 min, 10 μL of phage suspension (in dilutions to deliver MOIs of 1:1, 10:1 or 100:1 [pfu/CFU]) or its vehicle buffer was injected in the opposite proleg. After these infection and treatment injections, worms were incubated in plastic Petri dishes at 37 °C and monitored for death over 4 days. Worms were considered dead when they displayed no movement in response to tactile stimuli. Two control groups were included in the experiment, an “untouched” control group that did not receive any injections, to ensure the health of the worms after shipping, and a PBS control group that was injected with PBS instead of bacteria, to control for any detrimental effects from injection.

### 4.9. Statistical Analysis

Kaplan-Meier survival curves were compared using the Log-rank (Mantel-Cox) test with Bonferroni’s correction for multiple comparisons. Significance was established at *p* < 0.05. Statistical analysis was done using GraphPad software (http://www.graphpad.com/quickcalcs/), accessed on May 2022.

### 4.10. Accession Numbers

GenBank accession numbers for all phages are listed in Table 2.

## Figures and Tables

**Figure 1 antibiotics-11-01659-f001:**
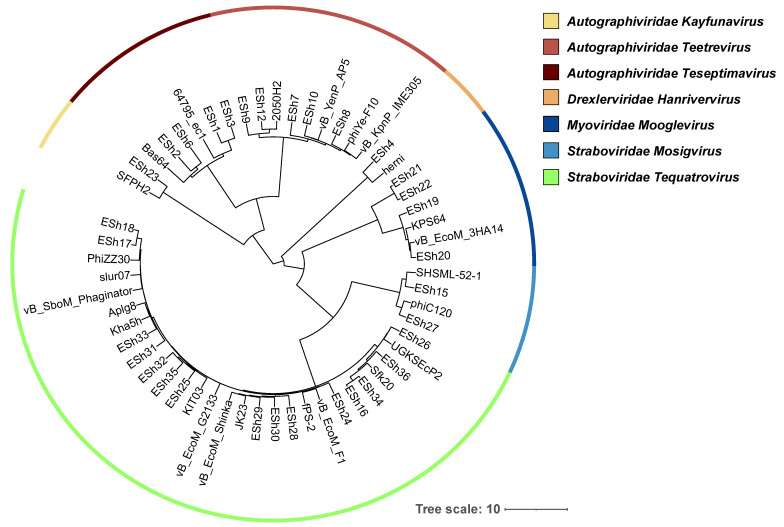
A whole-genome average nucleotide distance phylogenetic tree of the phages in this study. This tree was constructed for 57 total phage genomes from an ANI-based distance matrix calculated with MASH [35] using a sketch size of s = 5000, a k-mer size of k = 13 and GGRaSP [36] (see Section 4). Color strips denote genus-level taxonomic assignments (see key). The scale bar represents percent average nucleotide divergence. Genomes of the following phages were used as reference sequences: UGKSEcP2, *Shigella* phage Sfk20, *Escherichia* phage vB_EcoM_F1, *Yersinia* phage fPS-2, *Shigella* phage JK23, *Escherichia* phage vB_EcoM_Shinka, *Escherichia* phage vB_EcoM_G2133, *Escherichia* phage KIT03, Enterobacteria phage Kha5H, Enterobacteria phage Aplg8, *Shigella* phage vB_SboM_Phaginator, *Escherichia* phage slur07, *Serratia* phage PhiZZ30, *Shigella* phage SFPH2, *Escherichia* phage JeanTinguely strain Bas64, *Escherichia* phage 64795_ec1, *Serratia* phage 2050H2, *Yersinia* phage vB_YenP_AP5, *Yersinia* phage phiYe-F10, *Klebsiella* phage vB_KpnP_IME305, *Escherichia* phage herni, *Shigella* phage KPS64, *Escherichia* phage vB_EcoM_3HA14, *Shigella* phage SHSML-52-1, and *Escherichia* phage phiC120.

**Figure 2 antibiotics-11-01659-f002:**
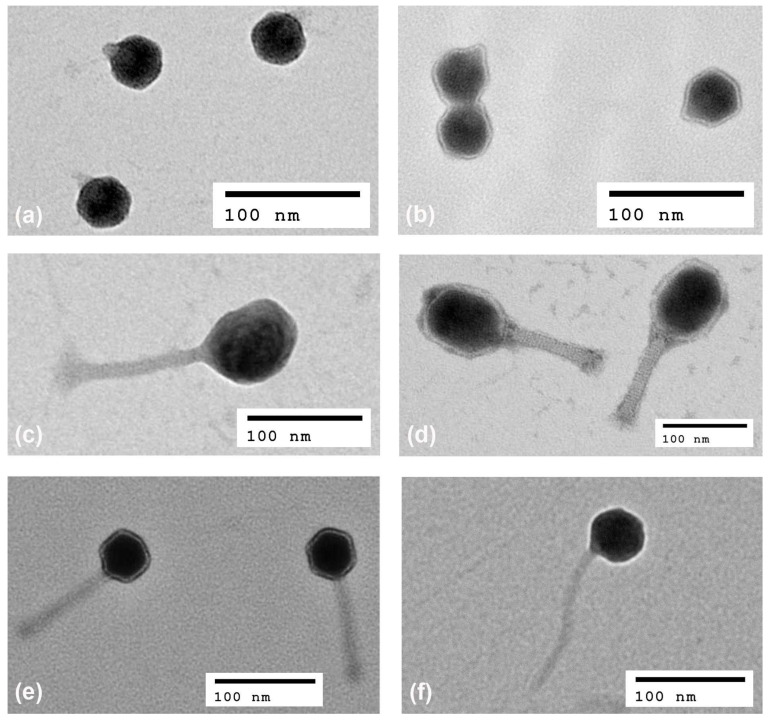
The morphology of *Shigella* phage particles via transmission electron microscopy. Podoviral morphology: (**a**) ESh9 (genus *Teetrevirus*); (**b**) ESh3 (genus *Teseptimavirus*). Phages with long contractile tails: (**c**) ESh15 (genus *Mosigvirus*); (**d**) ESh18 (genus *Tequatrovirus*); (**e**) ESh19 (genus *Mooglevirus*). Phage with long non-contractile tail: (**f**) ESh4 (genus *Hanrivervirus*).

**Figure 3 antibiotics-11-01659-f003:**
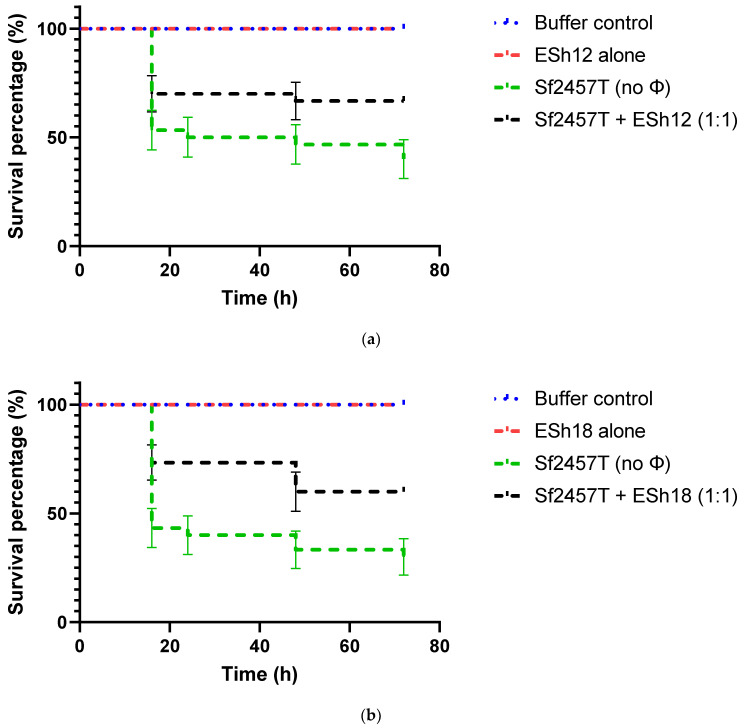
The survival of *G. mellonella* infected with *Shigella flexneri* strain 2457T: (**a**) treatment with phage ESh12 alone; (**b**) treatment with phage ESh18 alone; (**c**) treatment with phage ESh29 alone; (**d**) treatment with the 3-phage mixture. In each experiment the phage-treated group that received a single phage dose with multiplicity of infection (MOI) of 1:1 is shown. Controls used in each experiment: vehicle buffer alone, bacterial dose without phage treatment, phage treatment alone. Triplicates of experiments were conducted using ten worms per group. Pairwise comparisons of survival curves of treated versus untreated infected groups using the Mantel-Cox test: ESh12-treated (1:1) vs. untreated, *p* = 0.0439; ESh18-treated (1:1) vs. untreated, *p* = 0.0144; ESh29-treated (1:1) vs. untreated, *p* = 0.0003; Mix#4 (cocktail)-treated (1:1) vs. untreated, *p* = 0.0002.

**Table 3 antibiotics-11-01659-t003:** The lytic activity of 12 *Shigella* phages selected for use in prototype therapeutic mixtures.

Phage	Bacterial Host						Plaque Phenotype
	*S. flexneri* 2a	*S. flexneri* 3a	*S. flexneri* 5	*S. flexneri* 6	*S. sonnei*	*S. dysenteriae* 1	
ESh1	+	+	+	−	−	+	Large
ESh9	+	+	+	−	−	+	Large
ESh12	+	+	+	−	−	+	Very large
ESh16	+	+	+	−	+	+	Large turbid
ESh17	+	+	+	+	+	−	Small turbid
ESh18	+	+	+	+	+	−	Small turbid
ESh22	+	+	+	−	−	+	Large, halo
ESh27	+	+	+	−	+	+	Small turbid
ESh29	+	+	+	−	+	+	Small turbid
ESh31	+	+	+	−	+	+	Small clear
ESh33	+	+	+	+	+	−	Small clear
ESh35	+	+	+	−	−	+	Small clear

**Table 4 antibiotics-11-01659-t004:** A test of sterility conferred via lytic activity of prototype phage mixtures upon broth cultures of *S. flexneri* 2457T after 24 h of incubation.

Mixture	Phage Components			Sterility Test Result after 24 h Incubation
**#1**	ESh1	ESh18	ESh27	Sterile
**#2**	ESh12	ESh18	ESh27	Sterile
**#4**	ESh12	ESh18	ESh29	Sterile
**#15**	ESh1	ESh31	ESh33	Low secondary growth

**Table 5 antibiotics-11-01659-t005:** The host range of prototype phage cocktails against *Shigella* clinical isolates in the 95-strain diversity panel by species and serotype.

Bacterial Isolates		Lytic Activity of Phage Mixtures (%)			
	*n*	#1	#2	#4	#15
*S. sonnei*	7	100	100	100	100
*S. flexneri*	75	85.9	81.7	83.1	97.2
*S. dysenteriae* 1, 2	4	100	100	100	100
*S. dysenteriae* 9, 12	3	0	0	0	0
*S. boydii*	6	16.7	16.7	16.7	16.7
Overall *Shigella* collection	95	76.4	76.4	77.5	88.8

## Data Availability

Not applicable.

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
