# Peer review of "Design of a Bacteriophage Cocktail Active against Shigella Species and Testing of Its Therapeutic Potential in Galleria mellonella"

_antibiotics, 2022, doi:10.3390/antibiotics11111659_

Round 1

Reviewer 1 Report

The article by Filippov et al., describes the design and evaluation of phage cocktails for their therapeutic potential against Shigella species. 

This is an interesting, thorough, and well-written paper. There are some minor errors should be corrected before acceptance for publication.

1. Figure 2, Please add scale bars on the TEM images.

2. Please pay attention to the sub-headings and number them according to the journal guidelines.

3. Line 169: " a panel of 12 phages were selected" should be " a panel of 12 phages was selected"

4. Please ensure that the names of the Shigella strains used for phage enrichment are consistent (check 110-113 and 240-242).

5. Line 210: Please describe the P-value for the observed significant difference.

6. Please enrich discussion by adding more critical aspects associated with the use of 3-phage cocktail studied by authors.

Author Response

Thanks very much for your comments and suggestions.  Please find my point-by-point responses below:

  1. Figure 2, Please add scale bars on the TEM images.

>Thanks for this catch; we have added these to the revised figure.

  1. Please pay attention to the sub-headings and number them according to the journal guidelines.

>I have revised the subsections per the current journal manuscript template.

  1. Line 169: " a panel of 12 phages were selected" should be " a panel of 12 phages was selected"

>I have corrected this.

  1. Please ensure that the names of the Shigella strains used for phage enrichment are consistent (check 110-113 and 240-242).

>This information was consolidated into the Material & Methods section in a single sentence (there was a duplication in the manuscript, but the strains listed were the same), with references added per another reviewer.

  1. Line 210: Please describe the P-value for the observed significant difference.

>I added the p values to the figure legend.

  1. Please enrich discussion by adding more critical aspects associated with the use of 3-phage cocktail studied by authors.

>I am not entirely certain what the reviewer specifically would like to see added here.

Reviewer 2 Report

In the original article “Design of a bacteriophage cocktail active against Shigella species and testing of its therapeutic potential in Galleria mellonella” the authors have isolated 32 new bacteriophages infecting different Shigella species and serotypes. They have sequenced the phage genomes and characterized their safety and suitability for phage therapy purposes. Then, analyzed the infectivity of the newly isolated phages against 95 clinical Shigella strains. Finally, the authors prepared four prototype phage cocktails of which the most promising one was tested in a waxworm animal model. The manuscript is very well written and the topic is of great interest due to insufficient treatment and prophylaxis means against shigellosis.

Comments:

Major issues:

Although, the families Myoviridae, Podoviridae and Siphoviridae do not exist anymore (Turner, Kropinski, and Adriaenssens, 2021, A Roadmap for Genome-Based Phage Taxonomy, 10.3390/v13030506), authors use them throughout the manuscript. Thus, I suggest to the authors, please, carefully recheck the current classification of your new bacteriophages according to ICTV’s guidelines. 

Please, use same citation style in the text and in the reference list. Now, there are the authors’ names of the referred articles inside the text, but numeric system used in the reference list.

Minor issues:

Line 86: Consider adding other review(s) as well. For example, Tang et al. 2019, PeerJ.

Lines 92-95: Phrase “Optimization of the phage cocktail….” Is this statement opinion of Solodovnikov 1970 or authors own discussion? Please clarify. If own discussion, should this be in the discussion part?

Line 99:  “..bacterial resistance that emerges naturally” reference is missing.

Lines 110-113: This information should actually be in the Materials and Methods part. Also, I would like to have the references to these 8 different bacterial strains either in the text or in the Table 1.

Line 113: Please, take away word “see” before “Table 1”.

Line 115: Please take away word “see” before “Materials and Methods”.

Lines 125-128: Could you refer to the results (Table, Fig) or data not shown.

Table 1: Please do not use word “lab”, it is slang, but replace it with “laboratory”

Table 1: Please add references to the strains if they exist.

Table 1: Please bold also strain numbers after 50 and add something in the middle of the table to separate the two columns clearly.

Table 2: Please check the classification totally.

Table 2: Please open the abbreviation “nr”.

Table 2: Please change “General taxonomy for all phages” to f. ex. “General taxonomy for all phages isolated in this study”.

Figure 1: Please check the phylogeny.

Figure 1: Could you give more descriptive name to the Figure than “Phylogenetic tree”.

Figure 1: Please, use the full names of the phages in the figure caption. For example: F1 = vB_EcoM_F1

Line 152: Please remove the extra comma after fPS-2

Lines 156-162: Please, check the classifications and please add reference to the right Figure 2a, b, c…

Lines 161-162: “in agreement with sequencing results” What do you mean by this or how did you check this?

Figure 2: Please add scale bars to the figures.

Figure 2: Please check the classification/genus/family names.

Figure 2: Please add magnification and voltage in the figure caption.

Lines 169-171: Please could you explain more specifically, how did you select these 12 out of 32 phages here or in the Materials and Methods. Please don’t forget the inner references.

Line 172: Please, add “Table 4” in the end of the sentence.

Lines 174-176: “suggesting that this strain, which…” This is interesting, but consider moving this to the discussion section.

Table 4: Please add # before the number of the phage cocktails, since you have used it elsewhere

Table 4: Please consider replacing “mix” with word “mixture” or “cocktail” throughout the manuscript, since word “mix” is slang.

Line 188: Please indicate, who has isolated or where have you got these strains.

Line 189: Please correct the inner citation. I have not received Table S1. I think the reference should be to Table 1.

Line 193: Please add the percentage (93.7%) after 89/95 as previously. Please add “of all Shigella” before “strains”.

Table 5: Is it possible to add the number of each Shigella group (n) to an extra column.

Line 204: Please add “#4” in parenthesis after “ESh29”.

Line 211: Please add after Figure reference, whether you refer to a, b, c and/or d.

Lines 213: Please add Figure reference.

Line 214: Please, replace word “mix” with “cocktail”.

Line 216: Please add Figure reference.

Line 217: Please, add Figure reference.

Figure 3: Please, replace “Percent Survival” with “Survival rate (%)” or “Survival percentage (%)”.

Figure 3: Please check that the standard deviations can be seen in each line.

Figure 3: Would it be possible to use such lines that the figure could be read even in black/white format?

Figure 3: To the black line explanation, please, add “2457T” after “Sf”.

Figure 3: To the black line explanation, please add that 1:1 means cfu/pfu.

Line 221: Please, replace word “mix” with “cocktail (#4)”.

Line 224. What do you mean by challenge strain?

Discussion part in general very good, thank you!

Lines 349, 350, 356, and 370: Please, replace word “mix” with “cocktail”.

Lines 350, 355 and 358: Please, add figure references.

Line 360: Was this result mentioned in the Results section at all?

Line 383: Please add reference after “…lifestyle”

Line 387: Please replace word “pathogen” with “pathogenic”.

Line 391: Is there Table S1?

Line 393: Please add reference in the end of the sentence, Sambrook et al.?

Line 414: Please add “(MOI)” after “multiplicity of infection”

Line 418: Please replace word “precipitated” with “collected” or “concentrated”.

Line 419: Please remove the extra parenthesis.

Line 421: Please add the brand and manufacturer of the 0,22 µm membrane.

Line 425: Where does this endotoxin threshold come from? Please add reference.

Line 440-456: Please open abbreviations such as FLASH, ANI, MASH, UPGMA and APE R and add references to these programs/methods.

Line 463: Please add magnification used for the electron microscopy images.

Lines 482 and 484: Please add cfus and pfus rather than just microliters.

Lines 492-495: Please add references to the statistical methods used (when applicable), such as Kaplan-Meier, Mantel-cox test and Bonferroni’s correction.

Author Response

Thanks very much for your detailed review.  Please see below my point-by-point responses and actions taken in the revision (after the >):

Comments:

Major issues:

Although, the families Myoviridae, Podoviridae and Siphoviridae do not exist anymore (Turner, Kropinski, and Adriaenssens, 2021, A Roadmap for Genome-Based Phage Taxonomy, 10.3390/v13030506), authors use them throughout the manuscript. Thus, I suggest to the authors, please, carefully recheck the current classification of your new bacteriophages according to ICTV’s guidelines.

>I have updated the phage taxonomy to the current classification throughout the manuscript.  We also updated the phylogenetic tree figure to current classification.  Please note, however, that the taxonomists are constantly updating phage taxonomy at this time and it is very difficult to stay current with what they are doing.  I tried to update to current classifications throughout while still maintaining some reference to previous classifications that many in the field are still more familiar with.

Please, use same citation style in the text and in the reference list. Now, there are the authors’ names of the referred articles inside the text, but numeric system used in the reference list.

>I have changed this to the numeric scheme outlined in the Antibiotics journal manuscript template.

Minor issues:
Line 86: Consider adding other review(s) as well. For example, Tang et al. 2019, PeerJ.
> I added this as an additional reference.

Lines 92-95: Phrase “Optimization of the phage cocktail….” Is this statement opinion of Solodovnikov 1970 or authors own discussion? Please clarify. If own discussion, should this be in the discussion part?
>This was an opinion generally held, including by us, that I moved to the Discussion.

Line 99:  “..bacterial resistance that emerges naturally” reference is missing.
>I added a reference.

Lines 110-113: This information should actually be in the Materials and Methods part. Also, I would like to have the references to these 8 different bacterial strains either in the text or in the Table 1.
>I moved the duplicate line listing the strains to the Material & Methods (it was already there, actually), and I added references for all these strains in the text.

Line 113: Please, take away word “see” before “Table 1”.
>Done.

Line 115: Please take away word “see” before “Materials and Methods”.
>Done.

Lines 125-128: Could you refer to the results (Table, Fig) or data not shown.
>Done.

Table 1: Please do not use word “lab”, it is slang, but replace it with “laboratory”
>Done.

Table 1: Please add references to the strains if they exist.
>Done.

Table 1: Please bold also strain numbers after 50 and add something in the middle of the table to separate the two columns clearly.
>Done.

Table 2: Please check the classification totally.
>Done.

Table 2: Please open the abbreviation “nr”.
>I changed this the NCBI database; it is the nonredundant database there.

Table 2: Please change “General taxonomy for all phages” to f. ex. “General taxonomy for all phages isolated in this study”.
>This was/is actually intended to apply to all phages, not just those in the study/table.

Figure 1: Please check the phylogeny.
>Done and corrected.

Figure 1: Could you give more descriptive name to the Figure than “Phylogenetic tree”.
>Done; I added to the figure title to make it more descriptive.

Figure 1: Please, use the full names of the phages in the figure caption. For example: F1 = vB_EcoM_F1
>Done; I added the full names of all the phages to the legend.

Line 152: Please remove the extra comma after fPS-2
>Done.

Lines 156-162: Please, check the classifications and please add reference to the right Figure 2a, b, c…
>With respect, we think it is easier for the reader to find this in the figure legend rather than to list each panel of the figure in the text.

Lines 161-162: “in agreement with sequencing results” What do you mean by this or how did you check this?
>I edited the text to make the intent clearer.

Figure 2: Please add scale bars to the figures.
>Done; the figure was improved with the scale bars added to each panel.

Figure 2: Please check the classification/genus/family names.
>I reduced this to only genus names in the legend.

Figure 2: Please add magnification and voltage in the figure caption.
> Voltage is in the M&M. Scale bars obviate the need to list magnifications.

Lines 169-171: Please could you explain more specifically, how did you select these 12 out of 32 phages here or in the Materials and Methods. Please don’t forget the inner references.
>I added verbiage here to describe how the phages were selected.

Line 172: Please, add “Table 4” in the end of the sentence.
>It was already there, I think.  I have left it there.

Lines 174-176: “suggesting that this strain, which…” This is interesting, but consider moving this to the discussion section.
>I moved this to the Discussion.

Table 4: Please add # before the number of the phage cocktails, since you have used it elsewhere
>Done.

Table 4: Please consider replacing “mix” with word “mixture” or “cocktail” throughout the manuscript, since word “mix” is slang.
>Done.

Line 188: Please indicate, who has isolated or where have you got these strains.
>I have added some verbiage here to describe where the strain collection came from, and then more detailed information in the Material & Methods on the origin of the strains used for enrichment.

Line 189: Please correct the inner citation. I have not received Table S1. I think the reference should be to Table 1.
>Table S1 is correct.  Perhaps I didn’t submit that table properly so am adding it with the revisions.

Line 193: Please add the percentage (93.7%) after 89/95 as previously. Please add “of all Shigella” before “strains”.
>Done. However, adding “of all Shigella” is not accurate. I have edited it to make it clearer.

Table 5: Is it possible to add the number of each Shigella group (n) to an extra column.
>Yes, I added this.

Line 204: Please add “#4” in parenthesis after “ESh29”.
>I do not understand this suggestion; sorry.

Line 211: Please add after Figure reference, whether you refer to a, b, c and/or d.
>I am referring to the entire figure, all panels. I think it is correct as is.

Lines 213: Please add Figure reference.
> Done.  I added the figure panel for ESH29 alone and for the 3-phage mixture.

Line 214: Please, replace word “mix” with “cocktail”.
>Done.

Line 216: Please add Figure reference.
>Done.

Line 217: Please, add Figure reference.
>Done.

Figure 3: Please, replace “Percent Survival” with “Survival rate (%)” or “Survival percentage (%)”.
>Done.

Figure 3: Please check that the standard deviations can be seen in each line.
>I checked them; they are visible as long as the deviation is large enough.

Figure 3: Would it be possible to use such lines that the figure could be read even in black/white format?
>If the figure is changed to black and white it will not be possible to see the vehicle buffer and phage alone lines, since they are on top of each other.

Figure 3: To the black line explanation, please, add “2457T” after “Sf”.
>Done.

Figure 3: To the black line explanation, please add that 1:1 means cfu/pfu.
>Added in the Material & Methods section.

Line 221: Please, replace word “mix” with “cocktail (#4)”.
>Done.

Line 224. What do you mean by challenge strain?
 >I replaced this with correct and clearer language.

Discussion part in general very good, thank you!
 >Thanks!

Lines 349, 350, 356, and 370: Please, replace word “mix” with “cocktail”.
>Done

Lines 350, 355 and 358: Please, add figure references.
>For 350, we did not show those data and we noted that in the text already at the end of the sentence.  For lines 355 and 358, I added the figure reference.

Line 360: Was this result mentioned in the Results section at all?
>No, as we just said in the first part of the sentence, we did not show those data.

Line 383: Please add reference after “…lifestyle”
>I added a reference, though this is widely known for Shigella pathogens.

Line 387: Please replace word “pathogen” with “pathogenic”.
>I actually intended to use the word pathogen here and do not believe it is incorrect usage.

Line 391: Is there Table S1?
>Yes, per above I will make sure Table S1 is submitted with the revision.

Line 393: Please add reference in the end of the sentence, Sambrook et al.?
>I added this reference.

Line 414: Please add “(MOI)” after “multiplicity of infection”
>Done.

Line 418: Please replace word “precipitated” with “collected” or “concentrated”.
>Done.

Line 419: Please remove the extra parenthesis.
>Done.

Line 421: Please add the brand and manufacturer of the 0,22 µm membrane.
>Done.

Line 425: Where does this endotoxin threshold come from? Please add reference.
>I added verbiage on this to explain it.

Line 440-456: Please open abbreviations such as FLASH, ANI, MASH, UPGMA and APE R and add references to these programs/methods.
>I have done my best in this section to spell out all the abbreviations and add the references.

Line 463: Please add magnification used for the electron microscopy images.
>Again, with the scale bars added to the panels in the figure this is not necessary.

Lines 482 and 484: Please add cfus and pfus rather than just microliters.
>I added clarity on the CFU/mL in the suspension, but the volume injected is also critical so I retained that.  Hopefully my edit will clarify how we did this.  Different dilutions of the phage suspensions were used to obtain the different MOIs that were tested in the Galleria model, though we only showed data from the lowest MOI (1:1) that we tested.

Lines 492-495: Please add references to the statistical methods used (when applicable), such as Kaplan-Meier, Mantel-cox test and Bonferroni’s correction.
>These are common statistical methods we used that are not usually cited, even in Material & Methods sections.  However, we also provided the specific software package that we used to analyze the data (with link), so all the detail the reader could want could be obtained by going into that product description.

Reviewer 3 Report

The authors found 32 phage types that infect Shigella and performed infection characterization using 95 clinical isolates of Shigella. A phage cocktail was designed and the therapeutic efficacy was demonstrated using a wax moth larvae model. Further studies are needed to determine if they are also effective in treating Shigella infections in humans, but this study is comprehensive and the data are useful and should be published as is.

In Figure 2, scale bars should be shown for each picture.

Author Response

Thanks very much for your review.

In Figure 2, scale bars should be shown for each picture.
>Scale bars have been added to each panel of Figure 2.